# Knowledge, Readiness and Barriers of Street Food Hawkers to Support the Single-Use Plastic Reduction Program in Northeast Malaysia

**DOI:** 10.3390/ijerph19084507

**Published:** 2022-04-08

**Authors:** Nur Baizura Aini Abdullah, Nor Azwany Yaacob, Razan Ab Samat, Ahmad Filza Ismail

**Affiliations:** 1Department of Community Medicine, School of Medical Sciences, Health Campus, Universiti Sains Malaysia, Kubang Kerian 16150, Kelantan Darul Naim, Malaysia; drbaizura@gmail.com (N.B.A.A.); azwany@usm.my (N.A.Y.); 2Bachok District Health Office, Bachok 16300, Kelantan Darul Naim, Malaysia; razan@moh.gov.my

**Keywords:** single-use plastic, street food hawkers, pro-environmental behaviour, environmental knowledge, readiness, barriers

## Abstract

This study aimed to determine the knowledge, readiness, and barriers among street food hawkers to support the single-use plastic reduction program. A cross-sectional study was performed involving 440 night market food street hawkers from five districts in Kelantan, Malaysia selected through proportionate random sampling from 17 night market locations. The majority of the respondents had good knowledge level, 60% of respondents were ready to support this program, and 73% of them responded that barriers had low influence on them. Social media was the most popular information source utilized to obtain plastic usage information. Age, readiness to change, and significant barriers, were found to be associated with better knowledge. Male food hawkers and food hawkers that received information from social media and official sources were more ready to support single-use plastic reduction program. Proper strategies which incorporate more environmental knowledge, identify, and address the barriers may enhance the readiness to support this pro-environmental behaviour.

## 1. Introduction

Single-use plastics, also known as disposable plastics, are often used for packaging and include things intended to be used once and then discarded or recycled [1]. Food packaging is critical, and modern food systems would be unable to function without it. Single-use plastics are predominantly used for food packaging because they provide a barrier against light, protect against moisture, prevent contamination by a strong odour, and prolong food shelf-life [2]. Packaging accounts for most plastic consumption, accounting for 39.6% of total demand; one-third of worldwide plastic production is used for packaging [3]. The excessive production and consumption of plastic have severe consequences for the environment and human health.

Packaging is the most significant application area in the Malaysian plastic business. Malaysia accounts for the highest yearly per capita plastic packaging used in Southeast Asia, at 16.78 kg/person. By the end of 2020, the country’s total domestic plastic packaging usage is expected to exceed 523,000 metric tonnes [4]. Therefore, Malaysia is also at risk of having a solid waste problem due to excessive plastic use. Malaysia fell short of its 2020 goals of Malaysia’s National Strategic Plan (NSP) for Solid Waste Management set out in August 2005. The target was to divert 40% of garbage from landfills and increase recycling rates to 22%. The latest data indicate that almost 90% of solid waste was disposed of in sanitary landfills, with just 10.5% being recycled [5].

The Malaysian government introduced several initiatives to reduce single-use plastic, such as the No Plastic Bag Day (NPBD) Campaign in 2011 initiated by the Ministry of Domestic Trade Cooperatives and Consumerism (MDTCC). A plastic bag levy of 0.20 MYR per bag was added to strengthen the nationwide plastic ban [6]. However, small food businesses, such as night and public markets are exempted from the levy [7].

In 2020, a nationwide plastic straw banning program was introduced by implementing Malaysia’s 2018–2030 Roadmap toward zero single-use plastic consumption. This campaign created awareness of single-use plastic reduction among food hawkers; however, no enforcement has been made [8]. Although some restaurants have taken steps to use cardboard or paper containers for food delivery, the lids and coverings are still fabricated of plastic. Many food and beverage outlets also use plastic bags instead of paper bags for delivery as rainwater might soak into the paper bags during rains. There are no substitutes that reproduce plastic’s lightweight, transparent, waterproof, and resilient qualities in a way that properly seals food container packaging [9].

Environmental knowledge is the capacity to recognise various symbols, concepts, and behavioural patterns connected to environmental conservation [10]. People who are more knowledgeable of environmental concerns are more likely to behave pro-environmentally [11]. People who are more informed and convinced of their knowledge have a more positive attitude towards recycling behaviour [12]. Awareness of the negative impact of plastic use can be improved with additional knowledge about the correct food-grade plastics [13]. Proper channels, such as social media can disseminate better knowledge on single-use plastics as a severe environmental issue [14]. Sociodemographic factors, such as age, education level, gender and occupation may influence solid waste management behaviour [15].

Individuals’ readiness to be environmentally friendly can be understood by the belief that they can only commit something if they are “really ready” in terms of the ability or resources, such as education, time, and money. A lack of awareness about change and a lack of desire to support change are the top two reasons people resist change and why changes fail to achieve their objectives [16]. A recent survey on zero plastic use in Malaysia shows that Malaysians are aware of the environmental consequences of non-recyclable plastic products and are willing to take steps to mitigate the danger. They are, however, unwilling to invest more money to reduce the difficulties caused by excessive plastic use. Moreover, they believe the government should discover solutions to decrease the use of plastics. This finding supported a study by Hammami et al., who asserted that awareness of the harmful effects of plastics, did not affect usage behaviour [17].

Eliminating single-use plastics is challenging due to several barriers causing such initiatives to fail. Several factors can be used to categorise potential impediments: lack of clarity on the establishment of a plastic ban policy [18], lack of government initiatives to promote biodegradable single-use products, and no focused school education on plastic usage have been identified as barriers concerning top policymakers.

This may result in a lack of awareness of the single-use plastic ban, affecting hawkers and customers. Furthermore, the high cost of technology for single-use plastic alternatives [19], lack of other options for single-use plastic, and lack of manufacturing facilities [20] to meet the demand for biodegradable single-use products are also considered barriers. Fear of losing customers, growing customer reliance on plastics, a lack of other packaging options to improve product durability [21], and a lack of understanding of the harmful effects of plastic on humans and the environment [22] are barriers related to customers and hawkers.

It is necessary to investigate all components to evaluate food hawkers’ preparedness to support the single-use plastic reduction program.

Despite extensive awareness of the problem, the usage rate of plastic products is generally high [23]. It creates a dilemma since food hawkers also contribute to Malaysia’s massive plastic trash problem. While there is a sense of urgency and increased emphasis on plastics being an environmental concern, there is a data gap recording the public’s knowledge, readiness, and barriers to supporting plastic reduction programs, primarily among food hawkers. It is also critical to comprehend these three factors to solve this significant environmental issue.

Few studies in Malaysia have evaluated the thoroughness of general knowledge on single-use plastic containers concerning the type of plastic used as food containers, chemical hazards, the human and environmental impact of plastic use, and the law. One study examined the perception of plastic packaging used to pack hot foods among food hawkers at the night market in Kuala Selangor, Malaysia. This study evaluated the health hazards and practices related to plastic usage to pack hot foods [24]; it indicated a lack of readiness.

Another study in Penang, Malaysia, assessed the restaurants on environmentally friendly practices, drivers, and change barriers [25].

However, in this study, the barriers concept evaluations were conducted for independent casual upscale restaurant owners that offer fine dining with full table service rather than street food hawkers that include moving hawkers, stationary or temporary food services, unlike the present study.

A study in Malaysia involved a face-to-face interview of 350 households to assess the knowledge, awareness, and motivational factors for reducing plastic waste. This paper added information about knowledge, awareness, and motivational factors [12]; however, respondents in this study represented the consumers’ views on plastic waste and the key drivers that encouraged them to be involved in the “No Plastic Campaign” in Malaysia.

There are also doubts about whether the food hawkers are genuinely ready to change their plastic usage habits and support the single-use plastic reduction program in their food business; most studies assess consumers’ views only. So, this study aims to provide better insight into the knowledge, readiness, and barriers of street food hawkers to supporting the single-use plastic reduction program in Kelantan, Malaysia, and its associated factors. Countering the current unsustainable consumption pattern among food hawkers is unavoidably one of the difficulties that must be addressed to achieve sustainable development. The findings can be used to formulate an interventional program concerning behavioural changes in street food hawkers and monitoring and evaluation by the local authority. Moreover, policymakers can plan a better way to address the problem of excessive single-use plastic containers used by street food hawkers, shifting the current direction to a more sustainable pathway towards a cleaner and healthier environment.

## 2. Methodology

### 2.1. Sample Size and Sampling Method

This cross-sectional study involves five districts in Kelantan, Malaysia. Kelantan is situated northeast of peninsular Malaysia. Bachok, Tanah Merah, Jeli, Pasir Puteh, and Pasir Mas districts were chosen using simple random sampling, as shown in Figure 1.

Proportionate sampling was implemented to obtain the sample size of night market food hawkers from 17 night market locations to yield the 440 samples required for the study. The inclusion criteria were Malaysian street food hawkers 18 years of age or above, willing to participate in the study. They must be able to read and understand Malay.

### 2.2. Research Instrument and Data Collection

The survey was conducted using a newly developed and validated questionnaire named Street Food Hawkers Readiness Scale (SFH-RS) prepared in the Malay language. Questionnaire development and validation were conducted between December 2019 and March 2021, involving 660 food street hawkers from 22 night market locations in Kota Bharu District, Kelantan, Malaysia.

This psychometric tool contained three domains for assessing knowledge, readiness, and barriers of street food hawkers to support single-use plastic reduction programs. The SFH-RS scale was divided into four sections. Section 1 contains the demographic profile of the respondents. Section 2 consists of 22 items on the knowledge domain with “yes”, “no”, and “don’t know” responses. The knowledge domain assesses general knowledge on plastic usage, chemical components in plastic production, the health and environmental impact of plastic use, and laws concerning plastic use.

Section 3 discusses the readiness domain, comprising 15 items under 2 factors: total readiness for support and good motivation. Section 4 assesses the barrier domain, comprising 9 items under 2 factors: reluctant to change and comfortable using plastic food containers. A four-point Likert scale having “strongly disagree”, “disagree”, “agree”, and “strongly agree” options was used in Section 3 and Section 4.

The validation process involved content validity, face validity, and construct validity. All items in the knowledge, readiness, and barriers domains had a good Content Validation Index (CVI) of more than 0.83, indicating that the content in the instrument was relevant and representative of the targeted domain [26] 

Face Validation Index (FVI) for these items exceeded 0.83, indicating acceptable clarity and comprehensibility for questionnaire items [27].

All items in the knowledge domain showed good psychometric properties using a two-parameter logistic model of item response theory (2-PL IRT) analysis evaluating discrimination and the difficulty index. An Exploratory Factor Analysis (EFA) indicated that all items showed a factor loading of ≥0.4 [28], with Cronbach alpha between 0.7 and 0.8 [29]. In Confirmatory Factor Analysis (CFA), the final model of the SFH-RS tool, demonstrated acceptable factor loading with the Tucker–Lewis index = 0.906, comparative fit index = 0.916, root mean square error of approximation (RMSEA) = 0.056 [30,31], and composite reliability of Rykov rho between 0.757 and 0.887 [27].

The SFH-RS questionnaire was disseminated using a self-administered Google form as data were collected during the COVID-19 pandemic. After scanning the QR code and consenting to participate in the survey, respondents could access and complete the questionnaire. Data collection was performed during non-peak hours or after the night market closing hours. This time was chosen to prevent business interference and offer respondents ample time and freedom to complete the questionnaire.

### 2.3. Scoring and Interpretation

Responses to each item in the knowledge domain were “yes”, “no”, or “don’t know”. One mark was given for the correct answer “yes” and zero for the incorrect answers “no” or “don’t know”.

The total score for the knowledge domain was 22 marks. For the readiness and barriers domains, a four-point Likert scale with the scoring response of “strongly disagree” (1 point), “disagree” (2 points), “agree” (3 points), and “strongly agree” (4 points) was used to rate readiness (Section 3) and barriers (Section 4). The maximum readiness and barrier scores were 60 and 36, respectively.

Knowledge level on plastic was categorised as good if the average score percentage was above ≥70% and poor if the total score was ≤69%. Food hawkers are tagged as ready if the average score percentage was ≥80% and not ready if the average score percentage was ≤79%. The average barrier score percentage of ≥70% indicated a strong influence, while a score of ≤69% showed a low influence of barrier factors that prevent street hawkers from supporting the single-use plastic reduction program.

### 2.4. Statistical Analysis

Data were compiled in an MS Excel spreadsheet and analysed using SPSS version 26. Descriptive statistics were used to summarise respondents’ sociodemographic characteristics. Numerical data were presented as mean (SD) or median (IQR) based on normality distribution. Minimum and maximum scores were measured separately for each domain. Categorical data were presented as frequency (percentage). The knowledge, readiness, and barrier scores were calculated by summing the points, dividing by total points and multiplying by 100 to obtain a percentage score.

Simple and multiple logistic regressions models were utilised to identify the association between demographic factors and knowledge with the readiness of food hawkers to support the single-use plastic reduction program. Variables with a *p*-value < 0.25, or statistically significant factors from simple logistic regression analysis, were selected for analysis using the multiple logistic regression model. Variables with a *p*-value < 0.05 were regarded as significant in multiple logistic regression analysis.

The Pearson correlation analysis was used to determine the correlation between the barriers effect and the readiness level of food street hawkers to support the single-use plastic reduction program. The Pearson correlation coefficient (r) provides information on the strength and direction of the relationship. A strong correlation corresponds to |r| ≥ 0.5, moderate with |r| ≥ 0.3 and <0.5, weak correlation with the |r| < 0.3, and <0.1 is considered insignificant.

A correlation of 0.5 might be regarded as strong in social science studies but weak in physical science studies, where instrumentation is extremely precise [32].

## 3. Result

### 3.1. Sociodemographic Background of Respondents

The SFH-KRB was completed by 440 night market food hawkers in the Kota Bharu district. The mean (SD) age was 33.7 (10.1) years of age. The genders were evenly represented, with 213 (48.4%) females and 227 (51.6%) males. Most respondents (46.4%) had a secondary school or lower educational background, followed by a diploma or equivalent (31.6%), and 14.4% were from a degree or higher education category. The majority of them (63.4%) had business experience of more than 3 years. Social media (69.1%) was the most significant information source used for obtaining information on plastic use, followed by radio (48.2%) and television (47.3%). The sociodemographic profile of the respondents is summarised in Table 1.

### 3.2. Knowledge, Barriers, and Readiness to Change Scores of Night Market Street Food Hawkers to Support the Single-Use Plastic Reduction Program in Kelantan

Most (71.8%) respondents had good knowledge scores with 79.2 (17.8) mean (SD); 22.7% was the lowest score.

The majority (73%) of the food hawkers responded that barriers had a low influence concerning support for the single-use plastic reduction program, while 60% (264) were ready to support the program. The results are shown in Table 2.

### 3.3. Knowledge Level and Associated Factors Concerning Street Food Hawkers’ Support for the Single-Use Plastic Reduction Program

Table 3 shows that a distinct linkage exists between age and knowledge level. A one-year age increase corresponds to 0.952 odds of food hawkers having poorer knowledge about plastic use (95% CI: 0.931, 0.974, *p*-value < 0.001). Meanwhile, a food hawker ready to change has 3.271 odds of having better knowledge levels (95% CI: 2.046, 5.230, *p*-value < 0.001) compared with food hawkers who are not ready to change. Additionally, food hawkers with significant barrier levels had 3.577 times the odds of having better knowledge levels (95% CI: 2.204, 5.805, *p*-value < 0.001) than food hawkers with no significant barriers.

### 3.4. Readiness Level and Associated Factors for Street Food Hawkers to Support the Single-Use Plastic Reduction Program

Multiple logistic regression analysis indicates that male food hawkers had 1.706 times the odds compared with females (95% CI: 1.124, 2.590, *p*-value = 0.012) in supporting the plastic reduction program. Meanwhile, food hawkers who received information from social media and official sources had 2.914 times (95% CI: 1.852, 4.584, *p*-value < 0.001) and 2.269 times (95% CI: 1.343, 3.835, *p*-value = 0.002) the odds, respectively, to support single-use plastic reduction program. The results are outlined in Table 4.

### 3.5. Correlation between Total Knowledge and Barriers to the Readiness of Street Food Hawkers in Kelantan to Support the Single-Use Plastic Reduction Program

A Pearson correlation analysis was run to determine the strength and direction of the relationship between knowledge, barriers, and readiness of food hawkers to support the single-use plastic reduction program. There was a significant and direct moderate correlation between knowledge and readiness scores (r = 0.492, *p* < 0.001). Food hawkers with higher readiness scores also performed better in their knowledge scores. In contrast, the barrier score has a strong indirect correlation with knowledge scores (r = −0.503, *p* < 0.001). This study shows that hawkers with higher barrier scores have lower knowledge scores. Additionally, the barrier and readiness score have a significant negative moderate correlation (r = −0.479, *p* < 0.001). This research demonstrated that food hawkers with higher barrier score showed a lower readiness score. These results are summarised in Table 5.

## 4. Discussion

This study is based on assessing street food hawkers’ knowledge, readiness, and barriers to supporting the single-use plastic reduction program. This study used a 46-item SFH-KRB questionnaire. The SFH-KRB questionnaire is a validated and reliable self-administered psychometric tool in the Malay language. It has four sections: (1) sociodemographic background, (2) knowledge domain (22 items), (3) readiness domain (15 items), and (4) barriers domain (9 items).

### 4.1. Sociodemographic Background of Respondents

In this study, the respondents’ mean (SD) age was 33.7 (10.1) years of age, with equal numbers of female and male respondents. In our research, most respondents (46.4%) had secondary school education qualifications, followed by a diploma (31.6%), while 14.4% had a degree or master’s qualification. The majority of the respondents had more than 3 years of business experience. In a study done in Pakistan on the knowledge of food handling among food handlers, the mean (SD) age was 35.3 years of age (11.9), ranging from 15 to 70 years of age. This trend was observed in many countries where street vending is considered the second employment opportunity and particularly important for young and middle-aged men [33].

Findings from the current study aligned with a study done among Malaysian food handlers in 2016, where 31% of the food hawkers had at least secondary level education, followed by a diploma (21.5%) and bachelor’s or postgraduate degrees (22.2%). However, in that study, most respondents (58.9%) had a working experience of fewer than 3 years [34]. This study also showed that most respondents used social media to obtain information on plastic food packaging (69.1%).

These findings were also seen in a South African study, where 42% of the respondent’s obtained information on plastic food packaging from informal sources (e.g., friends, family, media, and the internet) [35]. A study in Kenya also showed similar numbers: social media (36%), TV (29%), and radio (15%) were the main media channels through, which the youth acquired knowledge about plastic’s impacts on the environment and human health [14].

Social media is a popular tool to obtain information about plastic containers for food; easy information searchability contributes to its popularity [36,37].

### 4.2. Knowledge Level of Street Food Hawkers and Associated Factors

The knowledge domain questionnaire captures detailed information on plastic food containers, such as items concerning general knowledge on plastic use, chemical hazards of plastic food containers, health and environmental impact, and environmental law related to plastic food containers. This study found that most respondents (71.8%) showed a good knowledge level, attaining a score of more than 70%. Several studies show similar results. A study in Ghana revealed that most women food vendors had good environmental knowledge. They agreed that waste must be appropriately disposed of, and improper waste disposal is associated with environmental dangers such as floods and unpleasant surroundings [38]. Additionally, most food vendors (72.5%) in Bangladesh had adequate knowledge about food safety as they scored ≥75% in the survey. The same study showed that 48.5% agreed that utilising paper and polythene food containers for food packaging is harmful [39]. A study conducted at the National University of Malaysia (UKM) revealed that the UKM community’s level of knowledge regarding polystyrene goods varies depending on the respondents’ occupation [40]. In a study done in Mangalore, India, 86.4% of respondents knew at least one health hazard related to plastic usage, and age was a significant factor concerning plastic ban programs [41]. In contrast, a study from Egypt showed that 83.9% of the food hawkers had poor knowledge of endocrine-disrupting chemicals (EDCs). Food handlers with a science study background have a better knowledge score than a non-science study background, 42.9% vs. 31.4%, respectively [13].

This current study demonstrated that age is the only significant sociodemographic factor, having adjusted odds of 0.952 (95%CI: 0.93 to 0.97, *p*-value < 0.001). Results show that a 10-year increase in respondent age corresponds to 9.5 times poorer knowledge about plastic use. Another study in China showed similar results where age was associated with increased environmental knowledge. The study showed that younger ages were associated with increased knowledge but decreased enjoyment of natural experiences and ecological concerns. This observation contrasts with adult residents, where increasing age was associated with less objective knowledge but greater appreciation and environmental concern [42].

The research findings by Shalinawati et al. (2016) were different, suggesting no link between knowledge and sociodemographic status among food handlers in Kuala Lumpur, Malaysia; age and education level had *p*-values of 0.193 and 0.905 [34]. Another study in Malaysia also showed that a higher level of education significantly contributed to better knowledge levels (*p* = 0.09), while age (*p* = 0.240), gender (*p* = 0.836) and greater experience (*p* = 0.566) were not associated with good knowledge levels on food handling [43].

The results found that environmental knowledge positively affected attitude, perceived price on influenced attitude, perceived value affecting attitude and impact of product appearance on attitude [44].

A study among rural residents in China found that environmental knowledge explains behavioural intention to change. The study noted that rural residents had a significant awareness of the environment and its effects on their lives [45]. Eze and Ndubisi conducted a study in Malaysia and concluded that knowledge of environmental issues positively influenced consumer intention and actual purchase of green products. Thus, ecological knowledge may influence individual decision-making, which may affect actual behaviour to support green product purchases [46]. Carmi et al. stated that environmental attitudes can be cultivated through information, and environmental attitudes can lead to pro-environmental conduct. Consequently, they noted that environmental knowledge had an indirect effect on environmentally friendly behaviours and that there are other possible mediators in a causal relationship between environmental knowledge and ecologically friendly actions [47].

Additionally, food hawkers with significant barrier levels had 3.6 times the odds of having better knowledge than food hawkers with no significant barriers. This result might have some cognitive dissonance between their cognitive views, attitude, and behaviour.

Oshikawa introduced the Cognitive Dissonance Theory, stating that individuals retain their cognitive view, past behaviour, attitude, and environments [48]. Even when individuals express very positive attitudes toward green products, they frequently exhibit incongruous behaviours and fail to purchase them. In our study, the excellent knowledge scores representing their views might contradict their readiness to support the single-use plastic reduction program, as with a high barrier score. Hence, they might be displeased or disturbed concerning support for the plastic ban program. Moreover, several barriers related to the food business prevent them from supporting the single-use plastic reduction program.

Similar observations are evidenced in another study in Malaysia concerning polystyrene use, where restaurant operators had the highest level of knowledge, attitude, and practice (KAP) compared with night market hawkers. However, their polystyrene use practices were poor [49]. The findings by Akehurst et al. were different; they showed that good environmental knowledge was associated with a lower gap between attitude and willingness to purchase green products.

Individuals’ willingness to buy green products increases when they perceive them as beneficial to human health and the environment, reducing the gap between attitude and purchase behaviour [50].

People are generally more sensitive to environmental issues, especially those close to them. Environmentally friendly practices can be cultivated among Malaysian food hawkers by providing adequate knowledge. Such practices can attract new customers and recycling reduces costs [51].

### 4.3. Readiness Levels of Street Food Hawkers to Support the Single-Use Reduction Program and Associated Factors

Our study showed a significant difference between male and female hawkers’ readiness levels. It was found that male food hawkers were 1.7 times more likely to support the plastic reduction program than females. The observations are similar to a study in Egypt where there was a significant difference between men and women in environmental concern levels. This finding suggested that men were more concerned about environmental issues, had more significant positive attitudes toward green purchases, and were more knowledgeable than women [52]. Another similar finding in China indicates that women are less active in supporting green products and less concerned about the environment than men, which is related to the higher educational levels of males in rural areas.

Migrants, males, bachelorhood, wealth, and education typically contribute to relatively higher levels of green product purchase support [45]. Overall, data indicate that women exhibit greener living and working habits than men. The first conflict is that women are financially constrained and bear greater responsibility but have a high social preference toward environmental protection. The gender wage gap is as large as 39% in the People’s Republic of China and more significant for older generations, partly explaining why women are less willing to consider financial burdens related to the environment. Overall, evidence shows that women have more environmentally friendly living and working practices than men. The first contradiction is that, despite their strong social preference for environmental protection, women are constrained due to financial dependence and additional responsibilities [53].

Meanwhile, food hawkers who receive information from social media and official sources had 2.9 times the odds to support the single-use plastic reduction program. A study in Egypt showed similar findings, with most participants (79.0%) obtaining knowledge about safe plastic containers using social media, primarily Facebook [54].

Social media plays an important role in shaping customer attitude and purchase intention toward green products [55]. Millions of people interact using web platforms every day. Social media operates at a massive scale and generates data at high speeds, serving as an easy-to-use information dissemination tool.

Recently, the number of individuals accessing social media and digital platforms for news and information increased dramatically. Documentary audio and video recordings are widely shared, exposing the numerous detrimental consequences of plastics on human health, the environment, and ecosystems. It was discovered that information interchange on social media platforms (e.g., YouTube, Twitter, and Facebook) impacts public opinion about the dangers of plastics [56]. However, the legitimacy of information disseminated on social media platforms is questionable compared with traditional news sources, such as news channels and newspapers; social media offers independence and freedom of expression [57].

Rapada et al. discovered that social media can improve the possibility of reducing plastic consumption with the likelihood of individual interest and ability to read the link associated with the post. This study shows that if information comes from well-researched studies, social media can significantly impact consumer behaviour regarding plastic consumption. This type of data can readily be turned into outcomes depending on own social media activity and contributions [58]. The combination of news media and social media provides a window into public expressions of social norms. Social media provides a platform for civic participation in a public and real-time setting where users can question the news media’s prevailing narrative [59].

Furthermore, official information sources were associated with 2.3 times higher hawkers’ readiness to support single-use plastic reduction programs. In Penang, Malaysia, a study showed that social influence significantly and positively contributed to green purchases by environmental green volunteers in Penang.

A social norm is an action an individual might perform considering a reference perspective [60]. The referent points can be friends, neighbours, profit or non-profit organisations, such as government or local municipal authorities. Official information source is a factor contributing to food hawkers’ readiness to support the plastic container reduction program for foodstuff. Information can be obtained through government policy instruments on plastic pollution at local, regional, and national levels. Voluntary information instruments are the official information sources for hawkers.

For hawkers, voluntary information sources include developing best practices, sharing data on new behavioural trends, guiding practices, and organising educational campaigns on green environmental behaviour [61].

In the Philippines, manufacturers were required to reduce the production of single-use plastic packaging at the national level; they were introduced to innovative alternative delivery systems or reusable packaging. The local government created an incentivised plastic waste collection program; people who managed plastic waste properly were motivated by offering Unilever products as rewards. Consequently, the program can develop a strong relationship between the local government unit, people, and private partners for environmental protection. To capture the target audience’s attention, they can utilise visual communication tools, such as posters, infographics, or movies that convey accurate information about single-use plastics. The use of visual communication aids information comprehension by the audience. It improves understanding [61].

### 4.4. Correlations between Knowledge, Readiness, and Barriers of Street Food Hawkers to Support the Single-Use Plastic Reduction Program

The Pearson correlation analysis was used to examine the relationship between the respondents’ knowledge, barriers, and readiness to support the single-use plastic reduction program. It was shown that knowledge score had a significant moderate and direct correlation with readiness score. In a study by Haron et al., knowledge significantly correlated with attitudes, behaviour, and participation among households in Malaysia. Respondents with good environmental knowledge tended to show a good environmental attitude and willingness to participate in sustainable consumption behaviour [62].

Additionally, another study in Malaysia proved that environmental knowledge directly, moderately, and positively influenced attitudes towards green brands, influencing green product purchase readiness [63]. It can be observed that the behaviour and readiness to change can be influenced by providing accurate knowledge on plastic, its impact, and the possible benefits of behavioural change [64].

Moreover, the study indicated a significant negative and strong correlation between knowledge and barrier scores. Better environmental knowledge and positive thoughts are insufficient to lead people to pro-environmental behaviour. Environmental knowledge and positive sentiments alone are inadequate for people to demonstrate environmentally responsible behaviour as barriers contribute to behavioural change [65]. Lack of understanding of green behaviours in the workplace is among the personal barriers affecting workplace behaviour. The workplace environment acts as a barrier factor and might encourage attitude, influencing behaviour.

Moreover, people might be affected by other social, cultural, and economic barriers that would determine how they choose to behave and perform in supporting environmentally friendly practices [66]. Due consideration of the barriers and increasing awareness might help employees engage in green practices to reduce the effects of environmental problems in organisations.

Moreover, this study demonstrated that barriers had a moderate negative relationship with readiness. Food hawkers ready to support plastic reduction programs are not expected to consider barriers as potential problems. A person who supports pro-environmental behaviour may exhibit pro-social behaviour, which is voluntary and intentional, either positive, negative or both, motivating and benefiting others.

People with pro-social behaviour will satisfy their personal needs and be more likely to support pro-environmental behaviour [11]. This behaviour was shown in a study involving green volunteers in Malaysia, where they were more action-oriented and readily participated in recycling activities and using green products. They were not interested in passive activities, such as attending talks or seminars, joining clubs or societies [67].

It was found that hawkers’ lack of readiness to support environmentally friendly food packaging was caused by environmental barriers and intrinsic factors [54]. Most food handlers in Penang, Malaysia, were concerned and informed about environmental issues, but the majority of them would not consider prioritising environmentally friendly business practices if the cost was too high, indicating a lack of economic benefit. Besides, restauranteurs’ barriers toward environmentally friendly practices included lack of diverse and competitively priced organic products, lack of societal demand, no trade pressure, old government policy, and weak enforcement [25].

## 5. Conclusions and Recommendations

This study was designed to assess the knowledge, readiness, and barriers of street food hawkers to support the single-use plastic reduction program and its associated factors. Most food hawkers had an appreciable knowledge score concerning plastic food containers and expressed readiness to support single-use plastic reduction programs. Considering that barriers had a low influence on pro-environmental behaviour support, there is hope that the single-use plastic reduction program can achieve its objectives.

Single-use plastic food container use among street food hawkers remains a significant public health issue. Considering extensive plastic food container usage and improper waste disposal, this study can fill the knowledge gap concerning readiness levels and barriers preventing hawkers from supporting green business behaviour. Besides, the study questionnaire can be used in other Malay-speaking countries. This study also reduces the information gap on the knowledge, readiness and barriers studied from the customer perspective.

The findings of this study can be implemented in numerous community settings to implement an effective single-use plastic reduction program. A proper educational and promotional program may enhance knowledge and understanding regarding single-use plastic food container usage. Besides, the local government can strengthen environmental laws by improving and establishing standards. Adequate enforcement can further reduce single-use plastic food containers among street food hawkers and improve the current solid waste management problem. Awareness activities to enhance readiness to support single-use plastic reduction programs can be planned to share this program’s advantages for food hawkers, the environment, and society. Moreover, local authorities can use study findings to promote the single-use reduction program by providing reasonably priced alternatives to single-use plastics.

This study has limitations because the data cannot be generalised to all of Malaysia; it might represent only the northeast part since it was confined to Kelantan. Future research should involve other regions in Malaysia and other races to improve data generalisability and outcome reliability.

## Figures and Tables

**Figure 1 ijerph-19-04507-f001:**
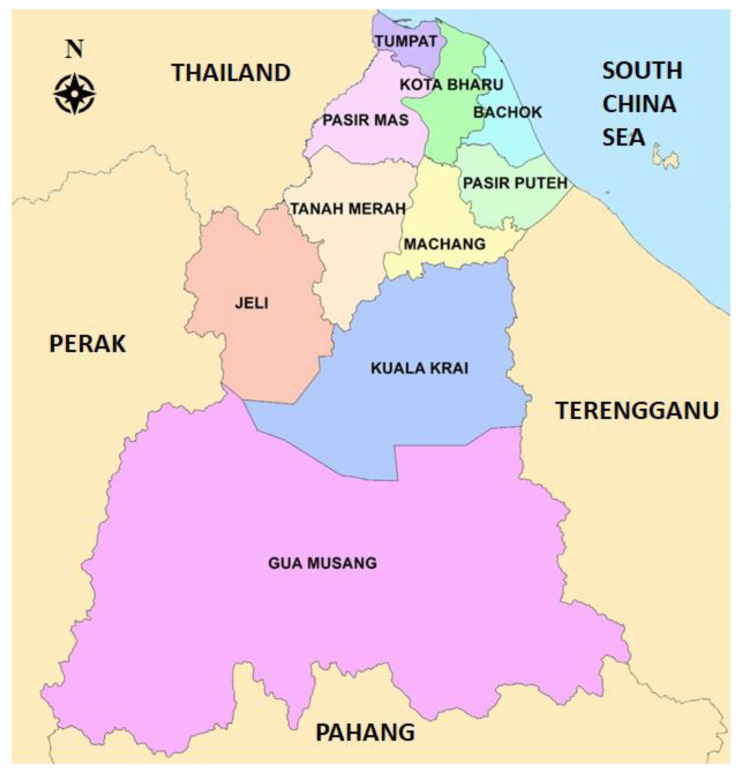
Map of Kelantan.

**Table 1 ijerph-19-04507-t001:** Night market street food hawkers’ sociodemographic background in the context of supporting the single-use plastic reduction program in Kelantan (*n* = 440).

Variables	Mean (SD)	*n* (%)
Age	33.7 (10.1)	
Gender		
Female		213 (48.4)
Male		227 (51.6)
Educational background		
No formal education		34 (7.7)
Secondary school or lower		204 (46.4)
Diploma or equivalent		139 (31.6)
Degree or equivalent		57 (13.0)
Master or equivalent or higher		6 (1.4)
Business experience		
Less than 3 years		161 (36.6)
More than 3 years		279 (63.4)
Information source		
Social media		304 (69.1)
Television		208 (47.3)
Newspaper		131 (29.8)
Official source		105 (23.9)
Advertisement		147 (33.4)
Radio		228 (48.2)

**Table 2 ijerph-19-04507-t002:** Knowledge, readiness, barrier scores, and score levels of night market street food hawkers to support the single-use plastic reduction program in Kelantan (*n* = 440).

Variables	Mean (SD)	Min–Max Score	Score Range	*n* (%)
Knowledge score	79.2 (17.8)	22.7–100.0	77.3	
Barrier’s score	61.6 (17.8)	27.8–100	72.2	
Readiness score	80.0 (15.2)	25.0–100.0	75.0	
Knowledge level				
Good				316 (71.8)
Poor				124 (28.2)
Barrier’s level				
Low influence				321 (73.0)
Strong influence				119 (27.0)
Readiness level				
Not ready				176 (40.0)
Ready				264 (60.0)

**Table 3 ijerph-19-04507-t003:** Multiple Logistic Regression based factors on knowledge scores of night market street food hawkers about supporting the single-use plastic reduction program (*n* = 440).

Variables	B	Adjusted OR (95% CI)	*p*-Value
Age	−0.049	0.952 (0.931, 0.974)	<0.001
Readiness level			
Not ready		1	
Ready	1.185	3.271 (2.046, 5.230)	<0.001
Barrier level			
Not significant		1	
Significant	1.125	3.577 (2.204, 5.805)	<0.001

Forward LR method was applied; no multicollinearity and no interaction; Hosmer Lemeshow test, *p*-value = 0.171. Classification table 77.7% correctly classified; area under the Receiver Operating Characteristics (ROCs) curve was 75.1%.

**Table 4 ijerph-19-04507-t004:** Multiple Logistic Regression based factors on the readiness level of night market street food hawkers about supporting the single-use plastic reduction program in Kelantan (*n* = 440).

Variables	B	Adjusted OR (95% CI)	*p*-Value
Gender			
Female		1	
Male	0.558	1.706 (1.124, 2.590)	0.012
Information source			
Social media	1.069	2.914 (1.852, 4.584)	<0.001
Official source	0.819	2.269 (1.343, 3.835)	0.002

Backward LR method was applied; no multicollinearity and no interaction; Hosmer Lemeshow test, *p*-value = 0.844; classification table 70.0% correctly classified; area under the Receiver Operating Characteristics (ROCs) curve was 71.9%.

**Table 5 ijerph-19-04507-t005:** Correlations between knowledge, barriers, and readiness scores of street food hawkers in Kelantan to support the single-use plastic reduction program (*n* = 440).

	Knowledge Score	Barrier Score	Readiness Scores
Knowledge score	-		
Barrier score	−0.503 *	-	
Readiness score	0.492 *	−0.479 *	-

* *p*-value < 0.001.

## Data Availability

Not applicable.

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
