# Peer review of "Knowledge, Readiness and Barriers of Street Food Hawkers to Support the Single-Use Plastic Reduction Program in Northeast Malaysia"

_ijerph, 2022, doi:10.3390/ijerph19084507_

Round 1

Reviewer 1 Report

This is good article and sound good to add new knowledge. Congratulation to all authors. This article can be more attractive by doing some improvement. I would suggest to the author to:

  1. Describe more on study gap compared to others articles.
  2. Improved the language by proofread.
  3. Take out all numbers and simbols in abstract (Line 16-21). 
  4. Explained more on implication of the study to the policy makers, enforcement body, respondents, hawkers, and environment.

Author Response

  1. Describe more on study gap compared to other articles.

Despite high awareness of the problem, usage rates of plastic products are generally high (Heidbreder et al., 2019). This created a dilemma since food hawkers also contributed to Malaysia's massive plastic trash problem. While there is a sense of urgency and increased emphasis on plastics as an environmental concern, there is a data gap recording the public's knowledge, readiness, and barriers to supporting plastic reduction programs, especially food hawkers.  It is also critical to comprehend these three factors to solve this major environmental issue.

 There is very minimal number of studies done in Malaysia to test the thorough general knowledge on single-use plastic containers covering the type of plastic to be used as food containers, chemical hazard, the human and environment impact of plastic use, and law involved. One study has been looking on the perception of plastic packaging usage to pack hot foods among food hawkers at night market in Kuala Selangor, Malaysia. In this study, health hazard and practice related to plastic usage to pack hot foods had been studied (Mat Issa, 2018) where there is lack in the rediness component. Another study done in Penang, Malaysia assessed the environmentally friendly practices among restaurant; drivers and barriers to change (Kasim and Ismail, 2012). However, in this study, the barriers concept measures were done among the independent casual upscale restaurants owners that offer fine dining with full table service rather than street food hawkers that include moving hawkers, stationary or temporary food services focused of this study. (Lines 104-122)

2. Take out numbers and symbols in abstract

Social media was the most popular information source utilized to get plastic usage information. Age, readiness to change, and significant barriers, were found to be associated with better knowledge. Male food hawkers, food hawkers that received information from social media and official source were more ready to support single-use plastic reduction program. Proper strategies which incorporate more environmental knowledge, identify, and address the barriers may enhance the readiness to support this pro- environmental behavior (Lines 16-21)

3.explained more on implications of the study to policy makers, enforcement body, respondents, hawkers, and environment.

Countering the current unsustainable consumption pattern among food hawkers is unavoidably one of the difficulties that must be addressed in order to achieve sustainable development. The findings could be used to formulate interventional program on the behavior changes of street food hawkers, interventional program, monitoring and evaluation by the local authority. Moreover, policy makers can plan a better policy to address the problem of excessive single-use plastic containers used by street food hawkers, shifting the current direction to a more sustainable pathway towards a cleaner and healtier environment.  (Lines 133-141)

4.The prime limitation of this article is INTRODUCTION. (Research Problem and its significant)

Already explained in items 1 and 3 above. Besides that we also mentioned this :

Another study done in Penang, Malaysia assessed the environmentally friendly practices among restaurant; drivers and barriers to change (Kasim and Ismail, 2012). However, in this study, the barriers concept measures were done among the independent casual upscale restaurants owners that offer fine dining with full table service rather than street food hawkers that include moving hawkers, stationary or temporary food services focused  of this study. A study done in Malaysia which involved selected 350 households to assess the knowledge, awareness and motivational factors towards plastic waste applying face-to-face interview. This paper added the knowledge, awareness and motovational factors information. However, respondents in this study represented the consumers views on plastic waste and the key drivers that encouraged them to be involved in “No Plastic Campaign” in Malaysia.

There is also a doubt whether the food hawkers are truly ready to change their plastic usage habit and switch to single-use plastic reduction program in their food business as more study done to assess the consumers views only. So, the purpose of this study was to give better insight on the knowledge, readiness, and barriers of street food hawkers to support single-use plastic reduction program in Kelantan, Malaysia, and its associated factors. Countering the current unsustainable consumption pattern among food hawkers is unavoidably one of the difficulties that must be addressed in order to achieve sustainable development.  

Reviewer 2 Report

  1. the prime limitation of this article is related to INTRODUCTION. The introduction of article should be sufficiently focused to the theme of this paper, i.e., RESEARCH PROBLEM, Both introducing of RP and it significance.

2 . To enrich the RESEARCH PROBLEM please refer to the below article,

The waste management of fruit and vegetable in wholesale markets: Intention and behavior analysis using path analysis

  1. Results and conclusion are okay.

Author Response

1. For sampling- provide a map would add information to the public
Map of Kelantan has been added to the manuscript.

2. it seems that the questionnaire is developed by research team. State this clearly, describe when it was developed, validated. 

The survey was conducted using a newly developed and validated questionnaire named SFH-RS (Street Food Hawkers Readiness Scale) that was prepared in Malay language. The development and validation processes of the questionnaire were conducted from December 2019 till March 2021 involving a total of 660 food street hawkers from 22 locations of night-market in Kota Bharu district, Kelantan Malaysia.

This psychometric tool contained 3 domains for the assessment of knowledge, readiness, and barriers of street food hawkers to support single-use plastic reduction program. The SFH-RS scale was divided into 4 sections. Section 1 contained the demographic profile of the respondents. Section 2 consisted of 22 items on the knowledge domain with the response option of “Yes”, “No” and “Don’t know”.  The knowledge domain assesses the general knowledge on plastic usage, chemical component in plastic production, the health and environmental impact of plastic usage and finally on the law related with plastic usage. Section 3 consisted of readiness domain which consisted of 15 items under 2 factors that were total readiness for support and good motivation. Section 4 assessed the barrier domain which consist of 9 items under 2 factors that were reluctant to change and comfortable using plastic food container. Four-point Likert scale was used in section 3 and 4, that were “strongly disagree”, “disagree”, “agree”, and “strongly agree”.

The validation process involved content validity, face validity and construct validity. All items in the 3 domains of knowledge, readiness and barriers had good Content Validation Index (CVI) of more than 0.83 and indicate that this instrument item content were relevant  and representative of the targeted domain (Yusoff, 2019).  Face Validation Index (FVI)  of items also attained more 0.83 which showed the items in this questionnaire good item clarity and comprehensibility (Yusof, 2019). All items in knowledge domain showed good psychometric properties through a two-parameter logistic model of item response theory (2-PL IRT) analysis from discrimination and difficulty index.  In Exploratory Factor Analysis (EFA) analysis, all items showed factor loading of   ≥ 0.4 (Hair et al., 2010) with Cronbach alpha between 0.7 to 0.8 (DeVellis and Thorpe, 2021). In Confirmatory Factor Analysis (CFA), the final model of SFH-RS tool demonstrated acceptable factor loading with Tucker-Lewis index = 0.906; comparative fit index = 0.916; and root mean square error of approximation (RMSEA) = 0.056 (Brown, 2015, Schreiber et al., 2006) with composite reliability of Rykov rho value between 0.757 to 0. 887 (Hair et al., 2010).

3. are there alternatives to plastic for hawkers? Is there a policy/regulation/method to access such alternatives?

Although some restaurants have taken steps to use cardboard or paper containers for food delivery, the lids and coverings are still made of plastic. Many food and beverage outlets also use plastic bags instead of paper bags for delivery as rainwater would soak into the paper bags during the rainy season. As for now, there is no way to reproduce plastic's lightweight, transparent, waterproof, and resilient qualities in a way that properly seals food container packaging (Yeoh, 2021).

4. some comments regarding studies carried out regarding the topic are not clearly discussed. Europe (Poland, Spain) and African (Egypt, Kenya) examples differ from the Asian models  

This is also evidenced in another study in Malaysia on practice in polystyrene use where food operators from restaurants had the highest level of knowledge, attitude and practice (KAP) compared to hawkers at night-markets. However, their practice in polystyrene usage was still poor (Abidin and Abedin, 2022). The finding of Akehurst et al. was different where it showed that less gap between attitude and willingness to purchase green product when they had good environmental knowledge. When an individual perceived green products to be beneficial to human health and  can protect environment, they would still willing to buy green product, thereby reducing the gap between attitude and purchase behavior [(Akehurst et al., 2012)]. People are generally more sensitive to the environmental issue especially those close to them. Environmentally friendly practice among food hawkers in Malaysia can be cultivated through good knowledge and it can attract new customers in their business, while recycling can reduces hawkers’ cost (JSJPN, 2009) .

5. the Spanish examples refers to buyers (382-386).

The Kenyan example (line 495-500) are not clearly discussed. 

the Spanish examples refer to buyers- removed, corrected version in line 405-417 as in item 4 above

the Kenyan examples refer to buyers- removed, corrected version referring to Malaysian study is available in line 536-543.

Most of food handlers in Penang, Malaysia were concerned and informed about environment issues, but majority of them would not consider prioritizing environmentally practices in their business if they found the cost was too high and consider this practice as not economically beneficial to them. Besides that,  other barriers of restauranteurs towards environmentally friendly practice were lack of diverse and competitively priced organic product, lack of societal demand, no trade pressure, and old governmental and weak enforcement [(Kasim and Ismail, 2012)]. 

6. conclusions need rewriting

This study was designed to assess the knowledge, readiness, and barriers of street food hawkers to support the single-use plastic reduction program and its associated factors. With most of the food hawkers had good score on knowledge of plastic food container, ready to support single-use plastic reduction program and considering that barriers had low influence on their support to this pro-environmental behavior, there are hopes that the single-use plastic reduction program could achieve its objectives.

Single-use plastic food container usage among street food hawkers remain a significant issue in public health. In the middle of extensive plastic food container usage and improper waste disposal problem, this article could serve to fill the gap knowledge on the data on knowledge, readiness level and barriers involved that prevent hawkers from suppoting this green behavior business practice. Besides that, the questionnaire used in this study can be used in other country which use Malay as their spoken language. This study also add on the information gap on the knowledge, readiness and barriers that were studied from the customers’ views. The findings in this study can be implemented for numerous community setting of green behavior for more effective single-use plastic reduction program among hawkers. For the limitation of the study, the data were not generalizable to whole Malaysia and  might only represent the north-eastern part of Malaysia as it was confined to Kelantan state. Future research should  involve other part of Malaysia and other races to improve data generalizability and more reliable result.

Reviewer 3 Report

The article has merit, and the introduction makes it clear why the study is timely and useful. Therefore, it is worth additional effort, to add clarity and strengthen the results. Below are some suggestions on the needed interventions.

Some more info on the hawkers could add valuable information (e.g. Deciphering Food Hawkerpreneurship: Challenges and success factors in franchising street food businesses in Malaysia, Ka Leong Chong, Marcus Lee Stephenson https://doi.org/10.1177/1467358420926695, or some other source).

Further:

  • For the sampling – providing a map would add information to the public, where these regions are, what territory is covered by the research.
  • It seems that the questionnaire (Street Food Hawkers Knowledge, Readiness and Barriers) – is developed by the research team authoring the study. State this clearly, describe when it was developed and tested/validated, where is it available etc. Maybe adding it as an Appendix would prove useful.
  • Are there alternatives to plastic use for street hawkers? Is there a policy/regulation/method to access such alternatives? What is the “right plastic”?
  • Some comments regarding studies carried out regarding the topic are not clearly discussed. European (Poland, Spain) and African (Egypt, Kenya) examples differ from the Asian models. A stronger explanation is necessary how the cited research compares to the Malaysian case.
  • The Spanish example (lines 3382-386) refers to buyers, not to vendors. Why is it relevant in this context? The same is true for Kenya (lines 495-500). Consumers buy what is offered. Vendors chose packaging according to availability, price, profit. So, the discussed barriers relate to the existence/availability of choices, not only to readiness/intentions.
  • Conclusions and recommendations section needs rewriting. The considerations are formulated broadly, do not refer directly to the research and evoke topics not developed in the article (e.g. that “plastic food container usage among street food hawkers remain a significant issue in public health. Excessive plastic food container usage had causes significant solid waste problem and it also can affect human health “).
  • Verify the references: Example: in reference (10) the name of the journal is missing (Environmental Education Research)

Author Response

1. For the sampling – providing a map would add information to the public, where these regions are, what territory is covered by the research.

The map has been added to the manuscript.

2.  It seems that the questionnaire (Street Food Hawkers Knowledge, Readiness and Barriers) – is developed by the research team authoring the study. State this clearly, describe when it was developed and tested/validated, where is it available etc. Maybe adding it as an Appendix would prove useful.

It has been added to the manuscript.

The survey was conducted using a newly developed and validated questionnaire named SFH-RS (Street Food Hawkers Readiness Scale) that was prepared in Malay language. The development and validation processes of the questionnaire were conducted from December 2019 till March 2021 involving a total of 660 food street hawkers from 22 locations of night-market in Kota Bharu district, Kelantan Malaysia.

This psychometric tool contained 3 domains for the assessment of knowledge, readiness, and barriers of street food hawkers to support single-use plastic reduction program. The SFH-RS scale was divided into 4 sections. Section 1 contained the demographic profile of the respondents. Section 2 consisted of 22 items on the knowledge domain with the response option of “Yes”, “No” and “Don’t know”.  The knowledge domain assesses the general knowledge on plastic usage, chemical component in plastic production, the health and environmental impact of plastic usage and finally on the law related with plastic usage. Section 3 consisted of readiness domain which consisted of 15 items under 2 factors that were total readiness for support and good motivation. Section 4 assessed the barrier domain which consist of 9 items under 2 factors that were reluctant to change and comfortable using plastic food container. Four-point Likert scale was used in section 3 and 4, that were “strongly disagree”, “disagree”, “agree”, and “strongly agree”.

The validation process involved content validity, face validity and construct validity. All items in the 3 domains of knowledge, readiness and barriers had good Content Validation Index (CVI) of more than 0.83 and indicate that this instrument item content were relevant  and representative of the targeted domain (Yusoff, 2019).  Face Validation Index (FVI)  of items also attained more 0.83 which showed the items in this questionnaire good item clarity and comprehensibility (Yusof, 2019). All items in knowledge domain showed good psychometric properties through a two-parameter logistic model of item response theory (2-PL IRT) analysis from discrimination and difficulty index.  In Exploratory Factor Analysis (EFA) analysis, all items showed factor loading of   ≥ 0.4 (Hair et al., 2010) with Cronbach alpha between 0.7 to 0.8 (DeVellis and Thorpe, 2021). In Confirmatory Factor Analysis (CFA), the final model of SFH-RS tool demonstrated acceptable factor loading with Tucker-Lewis index = 0.906; comparative fit index = 0.916; and root mean square error of approximation (RMSEA) = 0.056 (Brown, 2015, Schreiber et al., 2006) with composite reliability of Rykov rho value between 0.757 to 0. 887 (Hair et al., 2010).

3. Are there alternatives to plastic use for street hawkers? Is there a policy/regulation/method to access such alternatives? What is the “right plastic”?

Although some restaurants have taken steps to use cardboard or paper containers for food delivery, the lids and coverings are still made of plastic. Many food and beverage outlets also use plastic bags instead of paper bags for delivery as rainwater would soak into the paper bags during the rainy season. As for now, there is no way to reproduce plastic's lightweight, transparent, waterproof, and resilient qualities in a way that properly seals food container packaging (Yeoh, 2021).

4. Some comments regarding studies carried out regarding the topic are not clearly discussed. European (Poland, Spain) and African (Egypt, Kenya) examples differ from the Asian models. A stronger explanation is necessary how the cited research compares to the Malaysian case

This is also evidenced in another study in Malaysia on practice in polystyrene use where food operators from restaurants had the highest level of knowledge, attitude and practice (KAP) compared to hawkers at night-markets. However, their practice in polystyrene usage was still poor (Abidin and Abedin, 2022). The finding of Akehurst et al. was different where it showed that less gap between attitude and willingness to purchase green product when they had good environmental knowledge. When an individual perceived green products to be beneficial to human health and  can protect environment, they would still willing to buy green product, thereby reducing the gap between attitude and purchase behavior [(Akehurst et al., 2012)]. People are generally more sensitive to the environmental issue especially those close to them. Environmentally friendly practice among food hawkers in Malaysia can be cultivated through good knowledge and it can attract new customers in their business, while recycling can reduce hawkers’ cost (JSJPN, 2009) .

5. The Spanish example (lines 3382-386) refers to buyers, not to vendors. Why is it relevant in this context? The same is true for Kenya (lines 495-500). Consumers buy what is offered. Vendors chose packaging according to availability, price, profit. So, the discussed barriers relate to the existence/availability of choices, not only to readiness/intentions.

the Spanish examples refer to buyers- removed, corrected version in line 405-417

the Kenyan examples refer to buyers- removed, corrected version referring to Malaysian study is available in line 536-543

6. Verify the references:

Correction done. 

Reviewer 4 Report

The article concerns an interesting issue taking into account a comprehensive assessment of the knowledge on plastic recycling among street food hawkers and problems that arise with it. This work is of interest in the field of single-use plastics, but it has a number of questions that arise after reading the article.

  1. The article needs a text editing.
  2. What is the novelty of the proposed study?
  3. In the introduction section some important legal aspects in the field of plastics recycling are missing.
  4. The conclusion section is too long. Please improve it.

Author Response

1. The article needs a text editing.

The editing will be done after the technical issues being corrected.

2. What is the novelty of the proposed study?

This study is the using the newly developed and validated Malay version questionnaire and the focus group was street food hawkers, which is considered as B40 in the community.

3. In the introduction section some important legal aspects in the field of plastics recycling are missing.

Recycling is not widely practiced in Malaysia, especially in the state of Kelantan, where there is no regulation on recycling is enacted. Therefore, it is not included in the manuscript. 

4. The conclusion section is too long. Please improve it.

It has been modified.

  1. Conclusion and recommendation

This study was designed to assess the knowledge, readiness, and barriers of street food hawkers to support the single-use plastic reduction program and its associated factors. With most of the food hawkers having a good score on knowledge of plastic food containers, ready to support single-use plastic reduction programs and considering that barriers had a low influence on their support to this pro-environmental behavior, there are hopes that the single-use plastic reduction program could achieve its objectives.

Single-use plastic food container usage among street food hawkers remains a significant issue in public health. In the middle of extensive plastic food container usage and improper waste disposal problem, this article could serve to fill the gap knowledge on the data on knowledge, readiness level and barriers involved that prevent hawkers from supporting this green behavior business practice. Besides that, the questionnaire used in this study can be used in other countries which use Malay as their spoken language. This study also adds to the information gap on the knowledge, readiness and barriers that were studied from the customers’ views. The findings in this study can be implemented for numerous community settings of green behavior for a more effective single-use plastic reduction program among hawkers. For the limitation of the study, the data were not generalizable to the whole Malaysia and might only represent the north-eastern part of Malaysia as it was confined to Kelantan state. Future research should involve other parts of Malaysia and other races to improve data generalizability and more reliable result.

Round 2

Reviewer 3 Report

I read the revised manuscript and it gained clarity and an improved structure presentation. The references list shows an extensive interest in mapping the field on the use of plastics for takeaway food. The method is appropriate and allows the understanding of the conducted research. The Discussion section is improved. While the overall quality of the paper is good, the Conclusions and recommendations section seems a little bit underdeveloped. By comparison to the introductory part and to the discussion of results, there is an expectation for stronger points made and recommendations leading to transform readiness into action. 

Author Response

Reviewer comments:

While the overall quality of the paper is good, the Conclusions and recommendations section seems a little bit underdeveloped. By comparison to the introductory part and to the discussion of results, there is an expectation for stronger points made and recommendations leading to transform readiness into action. 

Reply:

Conclusion and recommendation part has been revised as follows:

This study was designed to assess the knowledge, readiness, and barriers of street food hawkers to support the single-use plastic reduction program and its associated factors. With most of the food hawkers having a good score on knowledge of plastic food containers, ready to support single-use plastic reduction programs and considering that barriers had a low influence on their support to this pro-environmental behavior, there are hopes that the single-use plastic reduction program could achieve its objectives. Single-use plastic food container usage among street food hawkers remains a significant issue in public health. In the middle of extensive plastic food container usage and improper waste disposal problem, this article could serve to fill the gap knowledge on the data on knowledge, readiness level and barriers involved that prevent hawkers from supporting this green behavior business practice. Besides that, the questionnaire used in this study can be used in other countries which use Malay as their spoken language. This study also adds to the information gap on the knowledge, readiness and barriers that were studied from the customers’ views.

The findings in this study can be implemented in numerous community settings for a more effective single-use plastic reduction program. Knowledge and understanding regarding single-use plastic food container usage may be enhanced through proper educational and promotional program. Besides that, establishment and improvement of environmental laws can be strenghten by the local government. Adequate enforcement could further reduce the single-use plastic food containers usage among street food hawkers and further improve the current solid waste management problem. Awareness activities to enhance readiness to support single-use plastic reduction program can be planned with the intention to share the advantages of this program that can benefit not only the food hawkers, but the whole society and the environment. Moreover, local authorities can use the study findings to promote the single-use reduction program by providing good supply of alternative to single-use plastics with reasonable price.  For the limitation of the study, the data were not generalizable to the whole Malaysia and might only represent the north-eastern part of Malaysia as it was confined to Kelantan state. Future research should involve other parts of Malaysia and other races to improve data generalizability and to obtain more reliable results.